# Multi-Task Federated Split Learning Across Multi-Modal Data with Privacy Preservation

**DOI:** 10.3390/s25010233

**Published:** 2025-01-03

**Authors:** Yipeng Dong, Wei Luo, Xiangyang Wang, Lei Zhang, Lin Xu, Zehao Zhou, Lulu Wang

**Affiliations:** 1State Key Laboratory of Intelligent Vehicle Safety Technology, Chongqing 401133, China; 51215902087@stu.ecnu.edu.cn (Y.D.); luowei2@changan.com.cn (W.L.); wangxy4@changan.com.cn (X.W.); 52265902009@stu.ecnu.edu.cn (L.X.); luluwang@stu.ecnu.edu.cn (L.W.); 2Shanghai Key Laboratory of Trustworthy Computing, Software Engineering Institute, East China Normal University, Shanghai 200062, China; 51275902016@stu.ecnu.edu.cn

**Keywords:** federated learning, multi-task learning, data privacy, split learning, multi-modal data

## Abstract

With the advancement of federated learning (FL), there is a growing demand for schemes that support multi-task learning on multi-modal data while ensuring robust privacy protection, especially in applications like intelligent connected vehicles. Traditional FL schemes often struggle with the complexities introduced by multi-modal data and diverse task requirements, such as increased communication overhead and computational burdens. In this paper, we propose a novel privacy-preserving scheme for multi-task federated split learning across multi-modal data (MTFSLaMM). Our approach leverages the principles of split learning to partition models between clients and servers, employing a modular design that reduces computational demands on resource-constrained clients. To ensure data privacy, we integrate differential privacy to protect intermediate data and employ homomorphic encryption to safeguard client models. Additionally, our scheme employs an optimized attention mechanism guided by mutual information to achieve efficient multi-modal data fusion, maximizing information integration while minimizing computational overhead and preventing overfitting. Experimental results demonstrate the effectiveness of the proposed scheme in addressing the challenges of multi-modal data and multi-task learning while offering robust privacy protection, with MTFSLaMM achieving a 15.3% improvement in BLEU-4 and an 11.8% improvement in CIDEr scores compared with the baseline.

## 1. Introduction

Federated learning (FL) [1,2,3,4] has emerged as a promising framework for distributed training of machine learning models, particularly in scenarios with privacy constraints [5,6,7]. This paradigm involves multiple clients collaboratively training a machine learning model, known as the global model, without openly sharing their respective local datasets. The fundamental concept revolves around the iterative exchange of model updates rather than the transmission of the datasets from clients. This iterative approach supports the creation of a global model based on the model updates while ensuring the confidentiality of each client’s dataset.

As the field of FL continues to evolve, a compelling extension known as multi-task FL across multi-modal data (MTFLaMM) has emerged [8,9,10]. This represents a significant advancement, involving the concurrent training of multiple machine learning models within the collaborative paradigm. This approach not only enhances the efficiency of model training but also safeguards the privacy of each client’s datasets. Moreover, through the incorporation of multi-modal data, such as text, images, and sensor data, MTFLaMM gains the capability to capture a richer and more nuanced representation of complex real-world scenarios. For instance, MTFLaMM is particularly advantageous for intelligent connected vehicles (ICVs) [11,12]. ICV systems require multi-task learning across diverse modalities, including visual data for object detection, textual data for navigation and communication, and sensor data for real-time traffic.

Handling multi-modal data and multi-task scenarios requires intricate model structures [13,14], leading to a notable increase in computing overhead. Clients may face challenges due to substantial computational requirements. In ICV systems, for example, the limited computing resources of vehicles make it challenging to carry out complex multi-modal, multi-task training locally. To address these issues, the adoption of split learning becomes a potential solution, alleviating training overhead on the client side by partitioning the training process. This approach empowers both the server and clients to independently oversee specific aspects of the training process, thereby alleviating the computational burden on individual clients. Thus, this paper focuses on multi-task federated split learning across multi-modal data (MTFSLaMM).

To design an efficient scheme for MTFSLaMM, a key challenge lies in the effective fusion of multi-modal data. Technologies such as transformers, knowledge graphs, and shared representation learning are commonly employed to address this challenge. Notably, transformers and their associated technologies have demonstrated outstanding results, among which the attention mechanism is one of the most popular technologies for transformers. However, when the attention mechanism is used with excessive layers in the model, it can lead to overfitting, reducing test performance and increasing computational costs, thus diminishing the model’s efficiency.

Another critical aspect requiring attention is the issue of data privacy within schemes for MTFSLaMM. Despite the growing significance of MTFSLaMM, it is noteworthy that the current research landscape largely lacks comprehensive investigations into the data privacy implications of these schemes. Specifically, while data privacy is well-acknowledged in FL, split learning, and their amalgamation, there remains a notable gap in the existing literature concerning data privacy specific to schemes for MTFSLaMM. This research scarcity underscores the urgent need for a thorough exploration of data privacy measures for MTFSLaMM schemes.

### 1.1. Related Work

In recent years, FL has seen significant advancements [15,16], but its integration with multi-task learning, multi-modal data, and their combined application remains relatively limited.

Several multi-task FL (MTFL) schemes have been proposed, making notable progress in addressing FL’s challenges. For example, Smith et al. [17] introduced MOCHA, which focuses on tackling communication costs and fault tolerance. Dinh et al. [18] proposed FedU, which extends MTFL by incorporating Laplacian regularization. Marfoq et al. [10] explored the use of personalized models, but faced challenges when clients have the data distribution of different modal data. Yu et al. [19] introduced the MTFL scheme tailored for the Internet of Things.

Integrating FL with multi-modal data is vital for leveraging diverse information. Frameworks like Zhao et al. [20], Peng et al. [21], and Zheng et al. [22] enhance feature extraction, resource distribution, and training stability in multi-modal FL but lack robust privacy protection. Privacy-focused methods, such as P2M2-CDR [23], using local differential privacy, and MuIMPC [24], integrating homomorphic encryption, improve security. But face limitations in scalability and multi-task applicability due to high overheads. Balancing privacy, efficiency, and adaptability in multi-modal FL remains a challenge.

As multi-modal data grow in importance, research integrating FL with both multi-modal and multi-task learning remains scarce, with notable efforts like [25] and FedMSplit [26]. The work in [25] adapts MTFL frameworks to multi-modal tasks such as vision and language by leveraging complementary information across modalities to enhance model performance. FedMSplit is essentially a scheme for MTFSLaMM. However, it lacks robust privacy-preserving techniques, highlighting the need for efficient, privacy-aware MTFSLaMM schemes to effectively address these challenges.

### 1.2. Contribution

FL schemes face notable challenges when applied to multi-task learning with multi-modal data. Existing MTFLaMM schemes partially address these challenges by extending FL to multi-task, multi-modal scenarios, but they often result in notable increases in computing and memory overheads. While split learning alleviates the computational burden by partitioning models between clients and servers, it fails to ensure robust privacy, leaving sensitive intermediate data shared during training and aggregation operations at the server susceptible to attacks.

To address these limitations, our main contribution is to propose a novel privacy-preserving scheme for MTFSLaMM. Concretely, our specific contributions are as follows:This novel MTFSLaMM scheme employs a modular design, partitioning the model into reusable components deployed on the server and task-specific modules on clients, effectively reducing communication overhead while utilizing server-side computational resources.To enable training on complex downstream tasks across diverse modalities, the scheme incorporates an adaptive aggregation method that dynamically integrates client models based on the type of downstream tasks.We address the challenge of effective multi-modal data fusion by employing an optimized attention mechanism. Specifically, we use mutual information as a measure to evaluate and refine the integration of multi-modal features. By systematically adjusting the number of attention layers, we maximize information fusion while minimizing computational overhead. This ensures that the model achieves high performance without overfitting, even with complex multi-modal datasets.To address the privacy vulnerabilities inherent in split learning, we integrate differential privacy to protect intermediate data and employ homomorphic encryption to secure client models during server-side aggregation.We validate the effectiveness of our approach on two multi-modal federated datasets under varying modality incongruity scenarios, demonstrating its ability to balance privacy, communication efficiency, and model performance.

The rest of the paper is organized as follows. Section 2 presents the research background. In Section 3, the privacy-preserving scheme for MTFSLaMM is proposed. The security of the scheme is analyzed in Section 4. The performance of the scheme is evaluated in Section 5. Finally, Section 6 shares our conclusions.

## 2. Background

### 2.1. System Architecture

We propose a novel scheme for MTFSLaMM, enabling clients to leverage data from various modalities for multiple tasks. Our system architecture, as illustrated in Figure 1, consists of clients and a central server.

Clients: Each client, typically resource-constrained entities such as intelligent vehicles in ICV systems, possesses datasets of various modalities. The clients aim to train a multi-task model using these datasets. In Figure 1, the multi-task model is partitioned into three components: model 1, model 2, and model 3. Models 1 and 2 are responsible for fusing multi-modal data. Specifically, model 1 integrates directly relevant information from different modal data features, while model 2 enhances the internal correlations among these features. Model 3 handles the downstream tasks. The client trains model 1 and sends the intermediate results to the server. After receiving processed results from the server, the client uses them as input to train model 3. When different clients wish to train the same downstream task using the same model structure, they homomorphically encrypt the downstream task model and send it to the server. The server leverages the additive properties of homomorphic encryption to aggregate models from different clients.Server: The server is responsible for training and aggregating parts of the model. It owns a portion of the model, receives activation data from the clients’ separation layer for training, and returns the output to the clients after adding perturbations for downstream task training. The server also handles aggregation: clients update the specific part of the model that requires aggregation, homomorphically encrypt it, and upload it to the server. After performing the aggregation, the server returns the result to the clients for decryption.

### 2.2. Design Goals

Given that models typically consist of numerous parameters, we adopt split learning to offload part of the model training to the central server, thereby reducing communication burdens in the MTFSLaMM scheme. However, since the central server could potentially compromise client data, our scheme must ensure the following properties:

Versatility: The scheme should handle multi-modal data, adapt to varying data distributions and patterns, and maintain robust performance across diverse domains. Our goal is to create a versatile system that facilitates effective knowledge transfer between tasks and modalities.

Efficiency: Due to the complexity of models that process multi-modal data, each round of FL incurs significant overhead. Therefore, enhancing training efficiency and minimizing communication overhead are crucial to optimizing performance.

Privacy: The scheme must ensure that clients’ local data and models remain private and are protected from potential attackers.

### 2.3. Differential Privacy

Differential privacy (DP), first proposed by Dwork [27], provides a robust criterion for ensuring privacy in distributed data processing systems. DP guarantees that the impact of altering a single record in the input dataset on the output is bounded within a specific threshold, making it infeasible for an attacker to infer changes caused by the inclusion or exclusion of a single record. We formally define DP as follows:

**Definition 1** (Differential Privacy). *A randomized mechanism M:X→R with domain X and range R satisfies (ϵ,δ)-DP if, for all measurable sets S⊆R and for any two neighboring datasets $D,D^{\prime }\in X$ (datasets differing in at most one record),*


(1)
$$\mathrm{Pr}\left[M\left(D\right)\in S\right]\leq e^{\epsilon }\mathrm{Pr}\left[M\left(D^{\prime }\right)\in S\right]+\delta .$$


In this definition, ϵ>0 is the privacy budget, controlling the distinguishability between outputs on neighboring datasets *D* and $D^{\prime }$. The parameter 0≤δ≤1 represents a small probability that the mechanism may fail to satisfy strict ϵ-differential privacy. Common DP mechanisms include the Laplace mechanism and the Gaussian mechanism, which add noise to the the output data. In this paper, we primarily apply the Gaussian mechanism to achieve the desired privacy guarantees.

### 2.4. Homomorphic Encryption

Homomorphic encryption (HE) [28] is an encryption method that allows specific algebraic operations to be performed directly on encrypted data, producing an encrypted result that, when decrypted, matches the result of operations performed on the plaintext data. In this paper, we use the Paillier homomorphic encryption scheme to protect the model. The Paillier cryptosystem consists of five algorithms: HE.KeyGen, HE.Enc, HE.Dec, HE.Add, and HE.Mul. These algorithms are defined as follows:HE.KeyGen(ℓ): Choose two large prime numbers *p* and *q* such that gcd(pq,(p−1)(q−1))=1, and ensure *p* and *q* have the same length. Compute n=pq and λ=lcm(p−1,q−1), where lcm denotes the least common multiple. Randomly select $g\in Z_{n^{2}}^{*}$ and compute $\mu ={\left(L\left(g^{\lambda }\;\mathrm{mod}\;n^{2}\right)\right)}^{-1}\;\mathrm{mod}\;n$, where $L\left(x\right)=\frac{x-1}{n}$. The public key is pk=(n,g) and the private key is sk=(λ,μ).HE.Enc(pk,m): Given a message $m\in Z_{n}$, generate a random number $r\in Z_{n}^{*}$ and compute the ciphertext $c=g^{m}r^{n}\;\mathrm{mod}\;n^{2}$.HE.Dec(sk,c): Given a ciphertext *c*, compute $m=L(c^{\lambda }\;\mathrm{mod}\;n^{2})\cdot \mu \;\mathrm{mod}\;n$.HE.Add($c_{1},c_{2}$): Given two ciphertexts $c_{1}$ and $c_{2}$ corresponding to plaintexts $m_{1}$ and $m_{2}\in Zn$, respectively, it computes $c_{1}c_{2}$
mod
$n^{2}$ = HE.Enc $(pk,\left(m_{1}+m_{2}\right)$
mod
*n*).HE.Mul(c,k): Given a ciphertext *c* corresponding to plaintext *m* and a scalar k∈Zn, it satisfies $c^{k}$
mod
$n^{2}$ = HE.Enc (pk,km
mod
n).

### 2.5. Attention Mechanisms

Attention mechanisms are essential components in modern machine learning. They allow models to focus on specific parts of the input when making predictions or generating outputs, rather than relying solely on a fixed-size representation of the entire input. We employ a multi-head attention mechanism to process multi-modal data. The multi-head attention (MHA) mechanism extends self-attention by utilizing multiple attention heads in parallel. Each head learns different weights, enabling the model to capture various relationships and patterns in the data. The outputs from all heads are concatenated and linearly transformed to produce the final output.

Given an input sequence of vectors $X=\{x_{1},x_{2},\ldots ,x_{n}\}$, the MHA mechanism is defined as follows:(2)$$Q_{i}=XW_{Q_{i}},\;K_{i}=XW_{K_{i}},\;V_{i}=XW_{V_{i}},$$
where $W_{Q_{i}}$, $W_{K_{i}}$, and $W_{V_{i}}$ are learnable weight matrices for the query, key, and value projections for the *i*-th head. The scaled dot-product attention for each head is computed as follow:(3)$$\mathrm{Attention}\left(Q_{i},K_{i},V_{i}\right)=\mathrm{softmax}\left(\frac{Q_{i}K_{i}^{⊤}}{\sqrt{d_{k}}}\right)V_{i},$$
where $d_{k}$ is the dimensionality of the key vectors. By using multiple attention heads, the model can focus on different parts of the input sequence simultaneously, capturing diverse patterns and relationships within the data. The final output is obtained by concatenating and linearly transforming the outputs from all attention heads.

## 3. Our Scheme

### 3.1. High-Level Description

Our proposed scheme consists of three main stages: initialization, model training, and aggregation.

Initialization: Each client selects pre-trained feature extraction models appropriate for their data modalities. Clients then jointly establish a public and private key pair (pk,sk) for homomorphic encryption. They also agree on the model to be trained, decide on the layers to be split, and send the portion of the model to be deployed on the server. The server deploys this received model segment.Model Training: Each client uses the pre-trained feature extraction models to extract multi-modal data features, forming feature matrices. The client inputs these feature matrices into model 1 for training. The trained output is then sent to the server with added DP perturbations. Upon receiving the client’s output, the server uses it as input to train model 2. The server then returns the trained result to the client, again protected by DP. Finally, each client uses the received result as input to train the corresponding downstream task in model 3.Aggregation: Clients homomorphically encrypt the updates of their locally trained downstream task model 3 and send them to the server. The server aggregates the model updates for the same downstream tasks, generates the corresponding global model updates, and returns the results to the respective clients. Clients decrypt the received results and update their local downstream models accordingly.

### 3.2. Initialization

In the initialization phase, each client selects appropriate pre-trained feature extraction models based on their data modalities. These pre-trained models do not add significant overhead to the system. For instance, if client $c_{i}$ possesses image and text data, they will locally deploy image and text feature extractors. Subsequently, the clients jointly establish a public-private key pair (pk,sk) for homomorphic encryption. Crucially, the clients also agree on the splitting of the training model *W*, deploying the client-side portion $W_{c}$ locally and uploading the server-side portion $W_{s}$ to the server. The server then deploys the received model.

### 3.3. Model Training

At this stage, clients and the server collaboratively process data features using models 1 and 2, as illustrated in Figure 2. Suppose client $c_{i}$ has three types of data—image, text, and audio—corresponding to visual, textual, and acoustic modalities. For instance, the IEMOCAP dataset [29] provides feature representations for these three modalities through feature extraction. We denote the extracted visual features as ${\vec{I}}_{i}=\left\{{\vec{i}}_{1},{\vec{i}}_{2},\cdots ,{\vec{i}}_{N}\right\}\in R^{N\times d}$, where *N* is the number of image candidate regions and *d* is the feature dimension. Similarly, the textual features are ${\vec{T}}_{i}=\left\{{\vec{\omega }}_{1},{\vec{\omega }}_{2},\cdots ,{\vec{\omega }}_{M}\right\}\in R^{M\times d}$, representing word embeddings for semantic concepts, and the acoustic features are ${\vec{V}}_{i}=\left\{{\vec{v}}_{1},{\vec{v}}_{2},\cdots ,{\vec{v}}_{S}\right\}\in R^{S\times d}$.

Before performing downstream tasks, client $c_{i}$ processes these features through three modules: the Integration Module on the client side, and the Generalization Module and Mapping Module on the server side.

Integration Module: This module, consisting of an MHA mechanism and a feed-forward network (FFN), fuses information from different modal data features. For example, if client $c_{i}$ needs to integrate three modalities, the module fuses the features accordingly. The execution process for client $c_{i}$ is simulated by the following equations:(4)$${\vec{I}}_{i}^{\prime }=FFN\left(MHA\left({\vec{V}}_{i},MHA\left({\vec{T}}_{i},{\vec{I}}_{i},{\vec{I}}_{i}\right),MHA\left({\vec{T}}_{i},{\vec{I}}_{i},{\vec{I}}_{i}\right)\right)\right),$$
(5)$${\vec{T}}_{i}^{\prime }=FFN\left(MHA\left({\vec{V}}_{i},MHA\left({\vec{I}}_{i},{\vec{T}}_{i},{\vec{T}}_{i}\right),MHA\left({\vec{I}}_{i},{\vec{T}}_{i},{\vec{T}}_{i}\right)\right)\right),$$
(6)$${\vec{V}}_{i}^{\prime }=FFN\left(MHA\left({\vec{T}}_{i},MHA\left({\vec{I}}_{i},{\vec{V}}_{i},{\vec{V}}_{i}\right),MHA\left({\vec{I}}_{i},{\vec{V}}_{i},{\vec{V}}_{i}\right)\right)\right).$$
Clients adjust the Integration Module based on their specific downstream tasks.

In the Integration Module, the attention mechanism fuses data features from different modalities. A single-layer attention mechanism may not effectively integrate information, potentially affecting model performance. Therefore, using multiple attention layers can deepen the connections between data and information. Measuring the degree of information fusion poses a challenge. Mutual Information (MI) is often used to quantify the amount of information one random variable contains about another and has been utilized to measure information changes in deep neural networks [30,31]. For datasets *X* and *Y*, the mutual information I(X,Y) is defined as follows:(7)I(X,Y)=H(X)−H(X|Y),
where H(X) is the entropy of *X* and H(X|Y) is the conditional entropy of *X* given *Y*. The effectiveness of information fusion is measured by calculating the MI between the original dataset and the fused data. Calculating MI on high-dimensional datasets is challenging due to difficulties in obtaining accurate conditional probability distributions. Therefore, estimation methods are used, which may introduce some differences from the true MI but generally follow the same trend. Monitoring these trends assists in configuring the model structure.

Generalization Module: Clients add differential privacy (DP) perturbations to the outputs of the Integration Module and send them to the server. The server receives each client’s results and processes them through the Generalization Module, which also includes an MHA and FFN. This module enhances the intrinsic correlations among features of the same modality, leading to higher-level representation information. For example, for image modality features, this module strengthens the connections between representations. The process for client $c_{i}$ is expressed as follows:(8)$${\vec{I}}_{i}^{*}=FFN\left(MHA\left({\vec{I}}_{i}^{\prime },{\vec{I}}_{i}^{\prime },{\vec{I}}_{i}^{\prime }\right)\right),$$
(9)$${\vec{T}}_{i}^{*}=FFN\left(MHA\left({\vec{T}}_{i}^{\prime },{\vec{T}}_{i}^{\prime },{\vec{T}}_{i}^{\prime }\right)\right),$$
(10)$${\vec{V}}_{i}^{*}=FFN\left(MHA\left({\vec{V}}_{i}^{\prime },{\vec{V}}_{i}^{\prime },{\vec{V}}_{i}^{\prime }\right)\right).$$

Mapping Module: This module maps the outputs of the Generalization Module to different task spaces, allowing the framework to adapt to various downstream tasks with different data spaces. Following the approach in [25], the mapping is defined as
(11)$$Mapping\left(x\right)=\tanh\left(xW_{m}+b_{m}\right)W_{mm}+b_{mm},$$
where $W_{m}$ and $W_{mm}$ are matrices for linear transformations, and $b_{m}$ and $b_{mm}$ are bias terms. For each downstream task, we apply the Mapping Module to transform the representations from the Generalization Module into the task space, specifically using LayerNorm(Mapping(${\vec{I}}_{i}^{*}$) + Mapping(${\vec{T}}_{i}^{*}$) + Mapping(${\vec{V}}_{i}^{*}$)), where LayerNorm denotes layer normalization. The Mapping Module can be tuned according to the downstream tasks of each client. Then, the server adds DP perturbations to the output of the Mapping Module and sends it to the corresponding client. The client then uses this result as input to train the downstream models.

### 3.4. Aggregation

In our framework, the Generalization Module and Mapping Module on the server process the outputs from all clients’ Integration Modules during training, eliminating the need for aggregation at those stages. However, aggregation becomes essential when multiple clients perform the same downstream tasks. By aggregating their models, the framework enhances generalization while preserving client-specific learning.

Figure 3 illustrates the aggregation process in our framework, where clients collaboratively contribute to training global models for shared downstream tasks. Each client trains its local models for specific tasks based on its available data. For example, Client 1 trains models for Tasks 1, 2, and 3, while Client 2 focuses on Tasks 1 and 2, and Client 3 trains models for Tasks 1 and 3. Once the local training is complete, clients encrypt their task-specific model updates using a shared public key based on Paillier homomorphic encryption. This encryption ensures that the server receives only encrypted data, preventing access to raw model parameters and preserving client privacy during transmission.

Upon receiving the encrypted updates, the server performs aggregation for each task using the additive properties of homomorphic encryption. This allows the server to compute the global model for a given task directly on the encrypted data without decryption. For instance, updates from all clients participating in Task 1 are aggregated to produce Global Model 1, while updates for Tasks 2 and 3 are similarly aggregated to create Global Models 2 and 3, respectively. This approach ensures that sensitive information from individual clients remains secure, even during collaborative computations on the server.

The aggregated global models are then sent back to the corresponding clients in their encrypted form. Clients decrypt these models using their private keys and integrate the updates into their local models for downstream tasks. By utilizing homomorphic encryption throughout the process, the framework maintains strict privacy guarantees while enabling effective multi-task learning. Figure 3 illustrates how the server aggregates encrypted updates and generates global models for shared tasks, ensuring efficiency and security in a federated learning environment. The overall training algorithm is outlined below (Algorithm 1).

**Algorithm 1** Multi-Modal and Multi-Task Federated Split Learning.
**Require:** 
Clients $C=\{C_{1},\cdots ,C_{n}\}$, where each $C_{i}$ holds its own dataset $D_{i}$ and downstream task models $M_{i}=\left\{m_{1},\cdots ,m_{j}\right\}$; public-private key pair (pk,sk) negotiated by all clients; *T* is the number of training rounds; *E* is the maximum number of local training epochs per round.  1:Each client $C_{i}$ selects pre-trained feature extraction models according to its data modality $D_{i}$.  2:**for** each round t=1,⋯,T **do**  3: **Clients**:  4: **for** each client $C_{i}\in C$ **do**  5:  Define modalities ${\vec{I}}_{i},{\vec{T}}_{i},{\vec{V}}_{i}$ based on actual data owned.  6:  Compute:  7:  
${\vec{I}}_{i}^{\prime }=Integration\left({\vec{I}}_{i}\right)$  8:  
${\vec{T}}_{i}^{\prime }=Integration\left({\vec{T}}_{i}\right)$  9:  
${\vec{V}}_{i}^{\prime }=Integration\left({\vec{V}}_{i}\right)$10:  Add DP perturbation:11:  
${\vec{I}}_{i}^{\prime }={\vec{I}}_{i}^{\prime }+{\mathrm{noise}}_{1}$12:  
${\vec{T}}_{i}^{\prime }={\vec{T}}_{i}^{\prime }+{\mathrm{noise}}_{2}$13:  
${\vec{V}}_{i}^{\prime }={\vec{V}}_{i}^{\prime }+{\mathrm{noise}}_{3}$14:  Send $\left\{{\vec{I}}_{i}^{\prime },{\vec{T}}_{i}^{\prime },{\vec{V}}_{i}^{\prime }\right\}$ to the server.15: **end for**16: **Server**:17: **for** each client $C_{i}$ **do**18:  
${\vec{I}}_{i}^{*}=Generalization\left({\vec{I}}_{i}^{\prime }\right)$19:  
${\vec{T}}_{i}^{*}=Generalization\left({\vec{T}}_{i}^{\prime }\right)$20:  
${\vec{V}}_{i}^{*}=Generalization\left({\vec{V}}_{i}^{\prime }\right)$21:  
$W=\mathrm{LayerNorm}(\mathrm{Mapping}\left({\vec{I}}_{i}^{*}\right)+\mathrm{Mapping}\left({\vec{T}}_{i}^{*}\right)+\mathrm{Mapping}\left({\vec{V}}_{i}^{*}\right))$22:  Add DP perturbation:23:  
$W=W+{\mathrm{noise}}_{4}$24:  Send *W* to client $C_{i}$.25: **end for**26: **Clients**:27: **for** each client $C_{i}\in C$ **do**28:  Use *W* as input for downstream training models $M_{i}$.29:  
$M_{i}^{\prime }=HE.\mathrm{Enc}\left(pk,M_{i}\right)$30:  Send $M_{i}^{\prime }$ to the server.31: **end for**32: **Server**:33: Aggregate encrypted models:34: 
$M^{\prime }=HE.\mathrm{Add}\left(M_{1}^{\prime },\cdots ,M_{n}^{\prime }\right)$35: Send $M^{\prime }$ to all clients.36: **Clients**:37: **for** each client $C_{i}\in C$ **do**38:  
$M^{*}=HE.\mathrm{Dec}\left(sk,M^{\prime }\right)$39:  Update local model $M_{i}$ with $M^{*}$.40: **end for**41:
**end for**



## 4. Security Analysis

In our proposed privacy-preserving scheme for MTFSLaMM, we recognize that while split learning provides some level of data privacy protection, it is insufficient to fully safeguard sensitive information. The scheme faces two primary privacy threats. First, during split learning, clients transmit intermediate data to the server, which may allow the server to infer private information from these data. Second, when client models are uploaded to the server for aggregation, the server may potentially extract sensitive information from these models.

To address these risks, we integrate a DP mechanism to secure the intermediate data transmitted over communication channels. During model training, the amount of intermediate data exchanged in split learning is relatively small. By applying the DP mechanism to these data, we effectively reduce the risk of information leakage while maintaining low communication overhead.

For client models requiring server-side aggregation, we utilize the Paillier homomorphic encryption scheme to ensure robust data privacy. This approach prevents the server from learning the contents of client models, even during aggregation, thereby mitigating the risk of privacy breaches by malicious servers.

By integrating differential privacy and homomorphic encryption, our scheme provides a robust and comprehensive solution to the privacy challenges inherent in MTFSLaMM.

### Privacy Protection Performance

To evaluate the privacy-preserving capabilities of our proposed framework, we employ membership inference attacks (MIAs) as a measure of vulnerability. MIAs are designed to determine whether a specific data point was included in a model’s training dataset, potentially revealing sensitive information. For example, in a medical dataset, an adversary might infer whether a person has a particular condition by identifying their inclusion in the training data.

We assess the framework’s resilience using two types of MIAs, which assume either black-box access or white-box access to the target model. Both approaches utilize a shadow auxiliary dataset to simulate adversarial scenarios:Black-Box Membership Inference (MIA$s_{\mathrm{Sh}}^{B}$). In this scenario, the adversary trains a shadow model using a dataset that mimics the target data distribution. The shadow model is queried with its training dataset to label samples as “in” (included in training) or “out” (excluded). Using this labeled dataset, a binary classifier is trained to predict the membership of samples in the target model.White-Box Membership Inference (MIA$s_{\mathrm{Sh}}^{W}$). Similar to the black-box attack, the adversary trains a shadow model to emulate the target model. However, with white-box access, the attacker exploits additional information, such as gradient updates, to extract enhanced features for membership prediction. This approach achieves higher accuracy by leveraging insights from stochastic gradient descent.

We conducted experiments using the MNIST dataset to measure the framework’s effectiveness. The dataset was divided into four subsets: target training, target testing, shadow training, and shadow testing, following a sample ratio of 2:1:2:1. Models were trained with varying privacy budgets, defined by the parameters ϵ and δ, while maintaining consistent training configurations as outlined earlier.

Table 1 summarizes the results of MIAs under varying privacy budgets (ϵ=2,4,6,8) and a non-private setting. The findings show that, under privacy-preserving conditions, the accuracy of MIAs remains close to random guessing, demonstrating the robustness of the proposed framework. For example, on the MNIST dataset, MIA accuracy under privacy-preserving configurations is approximately 0.50, even as ϵ increases, indicating effective mitigation of membership leakage. In contrast, under non-private conditions, MIA accuracy is significantly higher (e.g., 0.59 for MNIST), highlighting the vulnerability of unprotected models.

## 5. Performance Evaluation

We conduct an empirical study on two multi-modal joint datasets to explore two primary questions: (1) How does our scheme perform compared with baseline methods under different statistical and modality inconsistency settings? (2) What is the contribution of each module in our scheme to the overall performance? All experiments were conducted on a PC with an Intel Core i9—10850K CPU @ 3.60GHz, 32GB of memory—and an NVIDIA GeForce RTX 3060 GPU.

To simulate the behavior of multiple clients on a single PC, we employed Docker containers to create isolated environments for each client. This approach allows us to mimic the operations of independent clients, ensuring that each client operates with its own dataset and computational resources. By leveraging Docker’s lightweight virtualization, we efficiently manage the execution of concurrent client processes without significant overhead.

We select two multi-modal datasets to create our simulation environment:MSCOCO [32]: Developed by Microsoft, this dataset includes tasks such as detection, segmentation, and keypoint estimation. We leveraged it for the Visual Question Answering (VQA) task to assess each module’s overhead and our scheme’s overall accuracy.IEMOCAP [29]: This dataset is used for emotion recognition tasks and contains approximately 12 h of audiovisual data, including video, speech, facial motion capture, and text transcriptions. It is annotated with categorical labels like “angry”, “happy”, and “sad” by multiple annotators. Different clients randomly selected labels for classification. We employed corresponding feature extraction schemes for language, vision, and acoustic modalities.

Initially, we adjust the number of layers in the attention mechanism using MI to maximize information fusion without unnecessary computational overhead. We choose DeBERTa-v3-base as the text encoder and CLIP-ViT-base-patch16 as the image encoder for our experiments. Calculating MI accurately for high-dimensional data is challenging; therefore, we focus on adjusting the model structure based on the trend of MI changes rather than precise values. We use the CLUB method [33] proposed by Cheng et al. to estimate the MI between the initial data and the fused data. As shown in Figure 4, the MI value reaches its maximum when the number of attention layers is seven, indicating the optimal model configuration.

Next, we utilize the MSCOCO dataset for the VQA task. We set up ten clients, each holding a portion of the MSCOCO dataset. Each client aims to perform downstream VQA tasks. In addition, the server in our experimental setup is configured on the same PC but runs within a separate Docker container to ensure isolation from the client environment. The system is equipped with dedicated computing resources, including an Intel Xeon Silver 4114 CPU @ 2.20GHz, 64GB of RAM, and an NVIDIA TITAN V GPU, to efficiently handle the aggregation and training process. We use RCNN-based features extracted from Faster R-CNN [34], selecting 36 features for each image. The traditional federated learning framework serves as a baseline for comparison. We evaluate both the accuracy and the encryption overhead of our scheme. Table 2 compares the accuracy of the traditional federated learning framework and our proposed scheme. BLEU-4 and CIDEr are standard metrics used to evaluate the quality of generated text in tasks such as image captioning. BLEU-4 measures the correspondence between machine-generated text and reference text based on n-gram overlaps, with higher scores indicating better performance. CIDEr assesses the consensus between generated captions and multiple reference captions by considering term frequency-inverse document frequency weights, providing a more nuanced evaluation of caption quality. In terms of BLEU-4, compared with baseline, our scheme improves by 15.3%, which corresponds to an improvement of about 75.4% over baseline. This phenomenon is a direct indication of the higher accuracy of our scheme. In terms of CIDEr, compared with baseline, our scheme improves by 11.8%. This indicates that the captions generated using our scheme are closer to the reference captions, again demonstrating the higher accuracy of our scheme compared with baseline. This is attributed to the use of attention mechanisms before downstream training to enhance the integration of different modal data, thereby improving data quality.

While encryption and decryption processes can significantly impact overall time overhead, our scheme encrypts only a portion of the model, thereby reducing the overhead compared with traditional federated frameworks. Figure 5 illustrates the time required to encrypt and decrypt parameters of varying sizes.

We then use the IEMOCAP dataset to simulate sentiment classification training. Each client randomly selects one or more emotions for downstream classification tasks. Clients performing the same classification tasks have their downstream models aggregated. We conduct ablation experiments to evaluate the impact of different modules on model accuracy. Our scheme is compared with FedAvg [35], a fully global federated learning scheme, and Multi-FedAvg, which incorporates user multi-modal data. Table 3 presents the measurement results on the IEMOCAP dataset. Through the experimental results, we can find that it is reasonable that training multiple downstream task models is lower in accuracy compared with training a single downstream task model. When training a single downstream task model, the average accuracy of our scheme improves by 8.71% compared with FedAvg, and when training multiple downstream task models, the average accuracy of our scheme improves by 6.39% compared with Multi-FedAvg.

Our scheme protects intermediate data by adding DP during information transmission. However, adding DP can reduce model performance to some extent. To measure the impact of adding DP to transmitted information, we use the MNIST dataset. Figure 6 shows that while our scheme degrades model performance to some degree, the impact remains within an acceptable range.

Finally, we perform a functional comparison with other existing solutions. Based on the functional comparison summarized in Table 4, our proposed scheme demonstrates significant advantages over existing solutions in terms of reducing computational and communication overhead while ensuring privacy and security. Specifically, unlike the scheme in [25], which lacks privacy and security measures, and the scheme in [26], which fails to reduce computational and communication costs, our approach effectively addresses all three aspects. This comprehensive improvement highlights the practicality and robustness of our method in decentralized federated learning scenarios, where optimizing efficiency and safeguarding data privacy are critical requirements.

### Trade-Offs Between DP and Model Performance

The application of DP introduces noise into the system, which may result in some degradation in model performance. The acceptability of this degradation largely depends on the specific requirements and constraints of the application. For example, in fields such as healthcare or finance, where safeguarding data privacy is paramount, a higher degree of performance degradation may be considered acceptable in exchange for robust privacy guarantees. In contrast, applications with less stringent privacy requirements might demand stricter thresholds for performance retention.

In this study, we empirically analyzed the trade-offs between privacy and performance under different configurations of the DP mechanism. Specifically, we adopt a widely used privacy budget of ϵ=6, as it offers a practical balance between privacy preservation and model utility in many real-world scenarios. As presented in Table 2 and Figure 6, the degradation in model performance caused by DP noise is well-contained and remains within a practical range. Notably:IEMOCAP Dataset: Our proposed scheme consistently outperforms baseline methods such as FedAvg and Multi-FedAvg, even with DP noise applied, demonstrating its ability to maintain competitive performance in realistic federated learning environments.MNIST Dataset: The impact of DP noise on classification accuracy across varying training epochs (Figure 6) aligns with expected trends and remains within a range that does not compromise the utility of the model.

Although the precise definition of an “acceptable range” for performance degradation varies across applications, our experimental results provide meaningful benchmarks to guide practitioners. For privacy-sensitive use cases, we recommend adopting configurations with lower privacy budgets (ϵ) to strike an effective balance between privacy and utility. Conversely, for applications where performance is critical, DP parameters can be carefully adjusted to mitigate degradation while still ensuring adequate privacy protection.

## 6. Conclusions

We propose a privacy-preserving scheme for MTFSLaMM, designed to efficiently handle diverse data modalities while supporting multiple tasks. By leveraging split learning, the scheme reduces computational overhead for clients by partitioning the model and delegating part of the computation to the server. To ensure robust privacy protection, it integrates differential privacy to secure intermediate data exchanged during split learning and employs homomorphic encryption to safeguard client models during server-side aggregation. Experimental results demonstrate that the proposed scheme effectively enhances privacy while maintaining competitive performance in multi-modal and multi-task learning scenarios.

Our proposed MTFSLaMM scheme is particularly well-suited for ICV systems and other applications which require simultaneous execution of multiple tasks, such as object detection, traffic prediction, path planning, and driver behavior analysis, often relying on multi-modal data from diverse sensors like cameras, LiDAR, radar, and GPS. The modular split learning design not only reduces the computational burden on resource-constrained clients but also ensures that sensitive data remain protected through robust privacy mechanisms. Currently, the expensive communication overhead remains a bottleneck limiting the development of privacy-preserving, multi-task, multimodal federated learning, and future research could incorporate more efficient packet encryption or compressed transmission techniques to further reduce the communication overhead.

## Figures and Tables

**Figure 1 sensors-25-00233-f001:**
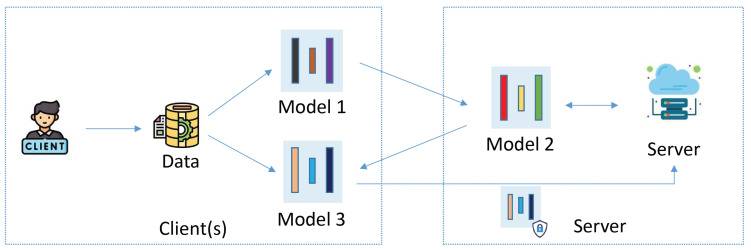
System architecture.

**Figure 2 sensors-25-00233-f002:**
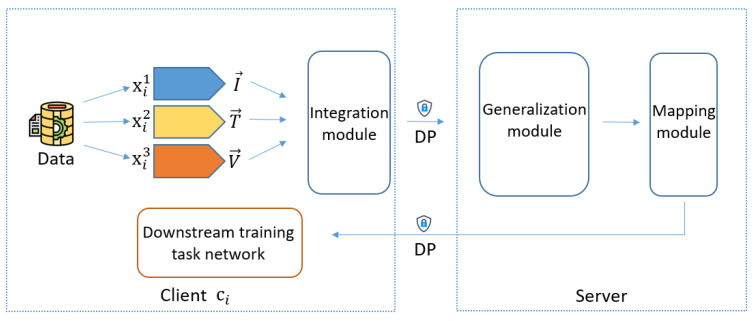
Model training.

**Figure 3 sensors-25-00233-f003:**
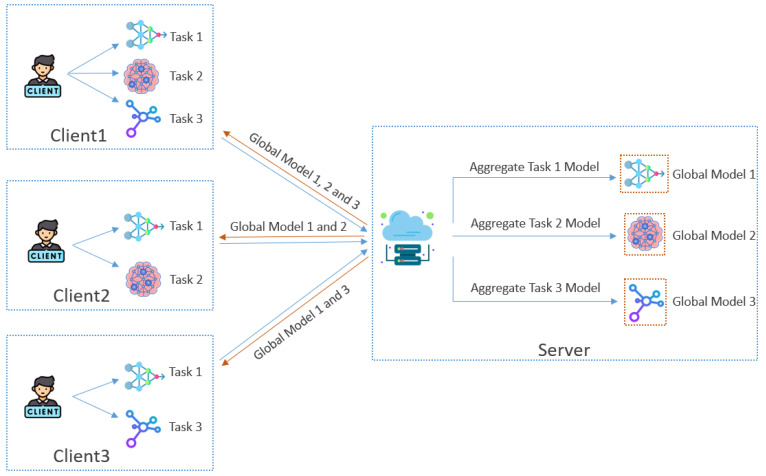
Model aggregation: Blue arrows depict the transmission of task-specific models from clients to the server for aggregation, while orange arrows illustrate the distribution of aggregated global models back to the clients.

**Figure 4 sensors-25-00233-f004:**
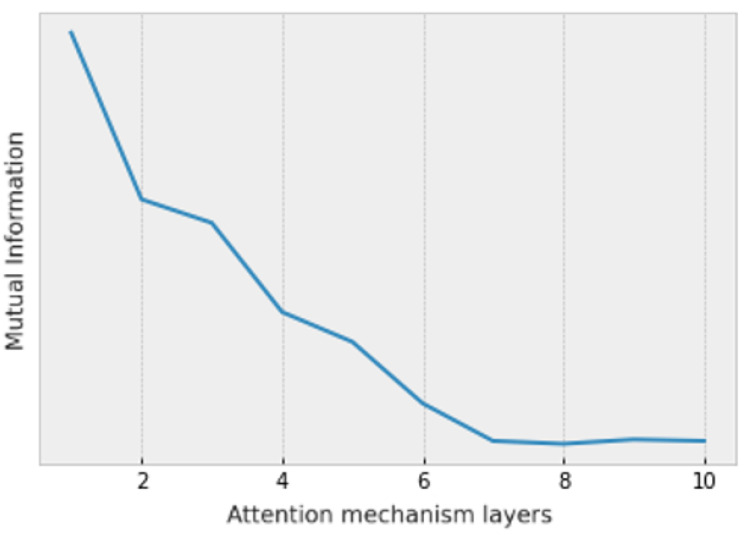
Impact of the number of attention layers on the mutual information (MI) between initial and fused data.

**Figure 5 sensors-25-00233-f005:**
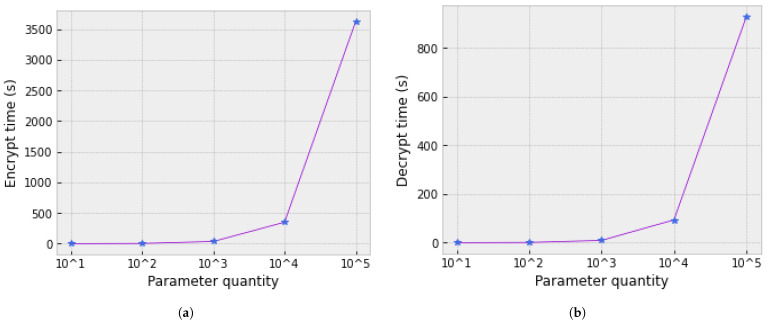
Encryption and decryption time as a function of parameter quantity, where the purple line represents the trend of time versus the number of parameters, and the blue star markers indicate the measured data points. (**a**) Encryption time vs. number of parameters. (**b**) Decryption time vs. number of parameters.

**Figure 6 sensors-25-00233-f006:**
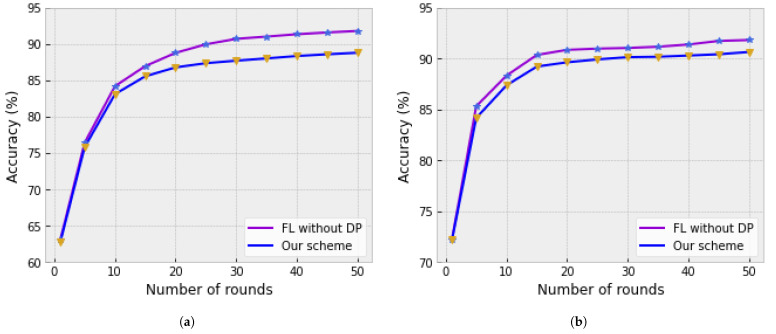
Classification accuracy comparison with different numbers of local training epochs, where the purple line represents the trend of accuracy versus the number of rounds, and the blue star markers indicate the measured data points. (L) per aggregation round. (**a**) Training with MNIST (L = 5). (**b**) Training with MNIST (L = 20).

**Table 1 sensors-25-00233-t001:** Accuracy of membership inference attacks on MNIST dataset.

Attack	ϵ=2	ϵ=4	ϵ=6	ϵ=8	Non-Private
**MNIST**					
MIA$s_{\mathrm{Sh}}^{B}$	0.491	0513	0.498	0.509	0.587
MIA$s_{\mathrm{Sh}}^{W}$	0.504	0.506	0.499	0.501	0.591

**Table 2 sensors-25-00233-t002:** Evaluation of the proposed scheme on the MSCOCO image captioning datasets. BLEU-4 and CIDEr are used as scoring indicators.

Methods	BLEU-4 (%)	CIDEr (%)
Baseline	20.3	105.8
Our scheme	35.6	117.6

**Table 3 sensors-25-00233-t003:** Average testing accuracy (%) on client testing data at global round T=10.

Methods	Single Downstream Task	Various Downstream Tasks
FedAvg	72.58	-
Multi-FedAvg	-	70.19
Our scheme	81.29	76.58

- indicates that there is no corresponding function.

**Table 4 sensors-25-00233-t004:** Functional comparison of different solutions, where Y indicates support for the functionality and N indicates the lack of support.

Schemes	Scheme in [25]	Scheme in [26]	Our Scheme
Reduce computational	Y	N	Y
Reduce communication	Y	N	Y
Privacy and security	N	N	Y

## Data Availability

No data were used to support this study.

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
