# Peer review of "Multi-Task Federated Split Learning Across Multi-Modal Data with Privacy Preservation"

_sensors, 2025, doi:10.3390/s25010233_

Round 1
Reviewer 1 Report
Comments and Suggestions for Authors
This paper proposes a novel privacy-preserving scheme for multi-task federated split learning across multi-modal data (MTFSLaMM).
There are some issues and concerns regarding the experimental setup and the description of the paper:
(1) The abstract mentions “Experimental results demonstrate the effectiveness of the proposed scheme” but does not briefly summarize the experimental results. It is recommended to briefly mention the experimental outcomes, such as “Experiments show a significant reduction in communication overhead.”
(2) The experiment mentions setting up 10 clients, each holding a portion of the MSCOCO dataset. However, since only one PC is used for the experiment, it is recommended to explain further how the behavior of multiple clients is simulated. For example, whether virtualization technologies, Docker containers, or other methods are used to simulate the operations of the clients.
(3) The experiment only mentions setting up 10 clients but does not mention the setup of the server. Additional details regarding the server configuration are recommended.
(4) Although the paper mentions that encryption and decryption processes can significantly impact the overall time overhead and that only a portion of the model is encrypted to reduce the overhead, it does not provide specific numbers or comparative results regarding the reduction in encryption and decryption overhead compared to traditional federated learning frameworks. It is recommended that a quantitative comparison with traditional methods be included to show the exact extent of the reduction in encryption and decryption overhead.
(5) It is recommended that more ablation experiments be added to explore the contribution of different modules to the final performance and include more experimental results.
(6) It is suggested that the Y-axis format of Figure 4 be changed.
(7) The experimental result metrics BLEU-4 and CIDEr in Table 1 should be explained.
(8) Figures 1, 2, 3, and 4 should be enlarged a bit.
(9) In Table 3, the functions of the scheme in [26] and the scheme in [27] are mentioned for comparison. It is recommended that some specific comparative experiments be added along with the corresponding results.
(10) The source code and benchmarks must be provided or released on an open-source repository (e.g., github.com ) to be available before the manuscript is accepted.
Comments on the Quality of English LanguageThe writing of this paper needs to be revised for scientific manuscripts. It needs to be written in better language and has some grammatical lapses and typographic errors.
Reviewer 2 Report
Comments and Suggestions for Authors
In this paper, the authors propose a privacy-preserving framework for multi-task federated split learning across multi-modal data. The framework combines differential privacy and homomorphic encryption to protect data and models while aiming to reduce computational overhead on resource-constrained clients (focus on ICV applications). The experimental results are promising and demonstrate the high potential of this framework.
In my opinion, a few things need to be considered for this article to present a more balanced and comprehensive version :
- While the results discussed time overhead from encryption and decryption, a breakdown of computational costs for individual modules (such as attention mechanism, DP application) is missing. This criterion is crucial to determining the practical feasibility, especially for clients with limited resources (hardware).
- As shown in experiments, adding DP noise degrades model performance. however, the acceptable range for degradation is not well-defined, which should include more clarified and rigorous benchmarks.
- Lastly, the security analysis is descriptive and lacks formal proofs or adversarial simulations that demonstrate resistance to specific types of attacks (e.g., inference, model reconstruction).
Reviewer 3 Report
Comments and Suggestions for Authors
This paper proposes a privacy protection scheme for multi-task federated split learning across multi-modal data (MTFSLaMM), utilizing modular design, differential privacy to protect intermediate data, homomorphic encryption to protect client models, and an optimized attention mechanism guided by mutual information. The experiment verified the effectiveness and privacy protection capability of the proposed scheme. Please refer to the comments below for potential improvements:
1. In section 1.2, it is recommended to use an itemized approach to introduce the contributions of this article.
2. The red wavy line in Figure 2 should be able to be removed.
3. The way of sandwiching paragraphs like Table 2 and Figure 6 may affect readers' reading, and it is recommended to make some adjustments.
4. Differential privacy can indeed protect deep gradient leakage, but it comes at the cost of affecting accuracy. Suggest adding some content to discuss the effectiveness and rationality of the accuracy affected in application scenarios. Or discuss how to strike a balance between reducing accuracy and protecting privacy.
5. The homomorphic encryption method will have an impact on communication time. Suggest adding experimental results in the experimental analysis section to verify the communication time advantage of the proposed method.
6. In Section 5, it is recommended to present more comparisons between the proposed solution and the baseline solution to demonstrate the advantages of this approach.
7. In Section 6, you can describe the possible application scenarios of this scheme and the next research direction.
Round 2
Reviewer 1 Report
Comments and Suggestions for Authors
All comments have been addressed, and then the manuscript can be accepted.
Reviewer 3 Report
Comments and Suggestions for Authors
Considering that the author has already answered all the questions I have raised, I agree to recommend the publication of this paper.